# Feasibility, Prediction and Association of Right Ventricular Free Wall Longitudinal Strain with 30-Day Mortality in Severe COVID-19 Pneumonia: A Prospective Study

**DOI:** 10.3390/jcm11133629

**Published:** 2022-06-23

**Authors:** Christophe Beyls, Tristan Ghesquières, Alexis Hermida, Thomas Booz, Maxime Crombet, Nicolas Martin, Pierre Huette, Vincent Jounieaux, Hervé Dupont, Osama Abou-Arab, Yazine Mahjoub

**Affiliations:** 1Department of Anesthesiology and Critical Care Medicine, Amiens University Hospital, F-80054 Amiens, France; tristan.ghesquieres@chu.amiens.fr (T.G.); booz.thomas@chu.amiens.fr (T.B.); crombet.maxime@chu.amiens.fr (M.C.); huette.pierre@chu.amiens.fr (P.H.); dupont.herve@chu.amiens.fr (H.D.); abouarab.osama@chu.amiens.fr (O.A.-A.); mahjoub.yazine@chu.amiens.fr (Y.M.); 2UR UPJV 7518 SSPC (Simplification of Care of Complex Surgical Patients) Research Unit, University of Picardie Jules Verne, F-80000 Amiens, France; 3Department of Cardiology, Amiens University Hospital, F-80054 Amiens, France; hermida.alexis@chu.amiens.fr (A.H.); martin.nicolas@chu.amiens.fr (N.M.); 4Respiratory Department, Amiens University Hospital, F-80054 Amiens, France; jounieaux.vincent@chu.amiens.fr

**Keywords:** right ventricle dysfunction, strain, COVID-19, speckle tracking, pneumonia

## Abstract

Introduction: Right ventricular (RV) systolic dysfunction (RVsD) is a common complication of coronavirus infection 2019 disease (COVID-19). The right ventricular free wall longitudinal strain parameter (RV-FWLS) is a powerful predictor of mortality. We explored the performance of RVsD parameters for predicting 30-day mortality and the association between RV-FWLS and 30-day mortality. Methods: COVID-19 patients hospitalized at Amiens University Hospital in the critical care unit with transthoracic echocardiography were included. We measured tricuspid annular plane systolic excursion (TAPSE), the RV S’ wave, RV fractional area change (RV-FAC), and RV-FWLS. The diagnostic performance of RVsD parameters as predictors for 30-day mortality was evaluated by the area under the receiver operating characteristic (ROC) curve (AUC). RVsD was defined by an RV-FWLS < 21% to explore the association between RVsD and 30-day mortality. Results: Of the 116 patients included, 20% (n = 23/116) died and 47 had a RVsD. ROC curve analysis showed that RV-FWLS failed to predict 30-day mortality, as did conventional RV parameters (all *p* > 0.05). TAPSE (21 (19–26) mm vs. 24 (21–27) mm; *p* = 0.024) and RV-FAC (40 (35–47)% vs. 47 (41–55)%; *p* = 0.006) were lowered in the RVsD group. In Cox analysis, RVsD was not associated with 30-day mortality (hazard ratio = 1.12, CI 95% (0.49–2.55), *p* = 0.78). Conclusion: In severe COVID-19 pneumonia, RV-FWLS was not associated with 30-day mortality.

## 1. Introduction

Right ventricular systolic dysfunction (RVsD) is a common feature [1] of severely ill coronavirus disease 2019 (COVID-19) patients with cardiac involvement, increasing morbidity and mortality [2]. In the intensive care unit (ICU), RVsD can be assessed by transthoracic echocardiography (TTE), a non-invasive, routine, simple bedside ultrasound technique [3]. TTE evaluation of the RV systolic function is crucial and recommended for the medical management of COVID-19 pneumonia [4]. However, the diagnosis of RVsD is challenging, requiring a multiparametric approach, and depends on the parameter used [5]. 

The RV free wall longitudinal strain (RV-FWLS), based on the bi-dimensional speckle tracking (2D-STE) method, is a prognostic, reliable, and accurate tool for the evaluation of RV systolic function in cardiovascular diseases [6]. RV-FWLS in a healthy individual is, on average, −30% [7], and a cut-off of −21% seems to be able to detect RVsD [5,6]. In COVID-19, RV-FWLS could be an interesting tool to identify patients with RVsD and increased mortality risk. RV-FWLS seemed to be reduced in COVID-19 patients compared to controls or reference values and represents a more powerful predictor of mortality than conventional RV parameters [8]. However, in COVID-19 pneumonia requiring mechanical ventilation, several studies found that RV-FWLS was not associated with the severity of the disease [9], cardiac biomarkers [10], and mortality [9,10,11,12]. Several known factors influencing the RV strain value (loadings conditions, heterogenous, and desynchronization of myocardial contractility) are indeed encountered in COVID-19 infection. The debate is still ongoing, and data are required to routinely use the RV-FWLS for mortality risk stratification in severe COVID-19 pneumonia admitted to the ICU. 

We aimed to explore the performance of RVsD parameters (TAPSE, RV-S’, RV-FAC and RV-FWLS) measured in TTE for predicting 30-day mortality in severe COVID-19 pneumonia patients. We also evaluated the feasibility of RV-FLWS and the association between RVsD defined by an RV-FLWS < 21% and 30-day mortality. 

## 2. Materials and Methods

### 2.1. Population

Adult patients admitted to our ICU for documented severe COVID-19 pneumonia with a TTE performed within 48 h of ICU admission were prospectively included in the study. Exclusion criteria included patients with atrial fibrillation during the TTE exam, permanent ventricular pacing, pregnancy, patients under extracorporeal membrane oxygenation (ECMO), and those with poor image quality for RV strain analysis. Patients were included on the day when TTE was performed. 

### 2.2. Ethics

This is an ancillary study of a prospective cohort study of patients with COVID-19 infection hospitalized in the ICU at Amiens University Hospital (NCT04354558). 

### 2.3. Data

Data from electronic data and medical reports were collected prospectively. A positive RT-PCR confirmed SARS-Cov2 infection on a nasopharyngeal swab or bronchoalveolar lavage on admission to our critical care unit. The simplified acute physiology score II evaluated the severity of illness upon ICU admission [13]. Vasopressor use was assessed by the SOFA cardiovascular (SOFA cv) score [14]. The different epidemic waves were characterized by three periods: (1) period A (between 28 February 2020 and 1 June 2020), (2) period B (between 1 September 2020 and 15 April 2020), (3) and period C (between 12 December 2020 and 15 March 2022). The 30-day and ICU all-cause mortality were collected.

### 2.4. TTE Measurement 

Trained operators performed TTE within 48 h of ICU admission. The TTE echocardiography protocol was used following the American Society of Echocardiography guidelines [15]. Echocardiographic images were obtained by a high-quality commercially available ultrasound system (CX 50, Philips Healthcare, Paris, France). All operators had a level III competence of general adult TTE [16]. 

RV size: We measured the RV basal, mid-cavity, and longitudinal linear dimensions in an RV-focused apical four-chamber view. RA volume was measured on the apical four-chamber view with 2D volumetric measurement based on tracings of the blood tissue interface and disc summation technique. RV dilatation was defined when the end-diastolic RV/LV area ratio was >0.6 in a four-chamber or subcostal view [5]. Acute cor pulmonale (ACP) was defined as RV dilatation (end-diastolic RV/LV area ratio > 0.6) associated with septal dyskinesia [17]. 

RV systolic function: RV conventionnal and 2D-strain parameters are summarized in Figure 1. We measured conventional RV parameters (TAPSE, RV-S’, and RV-FAC) according to international guidelines [18]: the RV-S’ wave was measured in the apical four-chamber view using Doppler tissue imaging mode (Figure 1A). TAPSE was measured using M-mode with a cursor placed at the junction of the lateral tricuspid leaflet and the RV-free wall (see Figure 1B). RV systolic and diastolic areas were measured in the apical four-chamber view in 2D mode. RV-FAC was calculated by subtracting the end-systolic area from the end-diastolic area and dividing this value by the end-diastolic area (see Figure 1C,D).

RV 2D-strain analysis: RV-FWLS was obtained using 2D AutoStrain software (AutoStrain, QLAB version 15, Philips Medical Systems, Andover, MA, USA) in an RV-focused four-chamber view at 50 to 70 frames/s. Fully automated RV-FWLS were measured by the software. We performed manual editing to fit RV myocardial wall thickness. The RV-FWLS was automatically calculated as the average of the three segments (basal, mid, and apical) of the RV-free wall (Figure 1E). The longitudinal strain was defined as the percentage of myocardial shortening relative to the original length and presented as a negative value, with a larger negative strain value reflecting better shortening. We used the absolute value of RV-FWLS for sampler interpretation. 

### 2.5. Statistical Analysis

Data are expressed as mean ± standard deviation (SD), median (interquartile range), or numbers (percentage), as appropriate. Firstly, a receiver-operating characteristic curve (ROC) was built to assess the prediction performance of RV-FWLS, TAPSE, RV-S’ wave, and RV-FAC for 30-day mortality. The area under ROC curves (AUC) of RV parameters was compared using Delong’s test. Secondly, the general population was dichotomized into two groups according to the presence of RVsD defined by an RV-FWLS < 21%. Variables were compared between groups (RVsD and non-RVsD group) using Mann–Whitney or Chi-square tests. Univariate and multivariate COX models were performed to evaluate independent factors associated with RV dysfunction. All factors with a *p* value < 0.05 in univariate analysis were included in the Cox model. The Kaplan–Meier method was used to plot the survival curves between the two groups, compared with the log-rank test. To assess intra-operator and inter-operator reproducibility for offline analysis, data of 10 patients were randomly selected and analyzed by the same operator and another operator with an interval of at least one week between the two analyses. The reproducibility of RV parameters was evaluated using the intraclass correlation coefficient (ICC). A statistical test was significant when the *p*-value was under 0.05. All *p* values are the results of 2-tailed tests. Statistical analyses were performed using SPSS software version 24 (IBM Corp, Armonk, NY, USA).

## 3. Results

Between 28 February 2020 and 1 March 2022, 404 consecutive patients were hospitalized in our ICU with COVID-19 pneumonia. Among the 232 patients with the inclusion criteria, 116 patients were excluded, notably 40 patients (35%) due to poor TTE image quality and 64 patients (55%) under ECMO (see Flow Chart, Figure 2). The feasibility of RV-FWLS was 74% (n = 116/156)**.** The study population was divided into two groups according to the presence of an RV dysfunction defined by an RV-FWLS < 21%. The study included one hundred and sixteen patients, with 69 patients in the no-RVsD group and 47 in the RVsD group. 

Demographics and biological and computed tomography data of the two groups are summarized in Table 1**.** There was no difference in age, SAPS II score, medical history, or biological and computed tomography data. In the RVsD group, there was a tendency toward a higher troponin level (27 (8–95) ng mL^−1^ vs. 16 (6–36) ng mL^−1^, *p* = 0.06). 

There was no difference between the two groups regarding vasopressor and invasive mechanical ventilation during the TTE exam (Table 2). For TTE parameters, there was no difference in RV size values (basal, mid cavity, and longitudinal). The left ventricular ejection fraction was similar between the two groups (*p* = 0.57). For RV systolic conventional parameters, patients in the RVsD group had a more impaired TAPSE (21 (19–25) mm vs. 24 (21–27) mm, *p* = 0.03) and RV-FAC (41 (34–47) % vs. 47 (41–54) %, *p* = 0.006). As was defined, RV-FWLS was lowered in the RVsD (17.5 (15.3–19.0) % vs. 26.7 (24.1–30.1) %, *p* = 0.0001). 

RV-FWLS and conventional RV parameters were included in a ROC analysis (Figure 3A) to estimate the probability of 30-day mortality and failed to identify patients with 30-day mortality. AUCs were near 0.5, and all *p* values were above 0.05. No cut-off values were determined. There was no difference in ICU complications or the 30-day mortality rate between the two groups (n = 13/69 vs. 10/47, *p* = 0.82; Table 2). 

The RVsD defined by the RV-FWLS < 21% (HR = 1.12, CI 95% (0.49–2.55), *p* = 0.78) was not associated with 30-day mortality in univariable Cox analysis (Table 3). In Cox multivariable analysis, age > 65 years (*p* = 0.001), acute cor pulmonale (*p* = 0.0001), and SAPS II (*p* = 0.005) remained independently associated with 30-day mortality. The analysis of survival curves by the Kaplan–Meier curves showed no significant differences between the two groups (log-rank test at 0.79, Figure 3B). The reproducibility of RV-FWLS was excellent, with an ICC of 0.84 CI 95% (0.57–0.96) and an ICC of 0.87 CI 95% (0.55–0.97) for the intra-operator reproducibility (Appendix A, Table A1). 

## 4. Discussion

The main findings of the present study can be summarized as follows: (1) RVsD defined by the RV-FWLS occurred in 40% of patients, (2) conventional and RV-FWLS parameters failed to predict 30-day mortality, (3) RV-FWLS was not associated with 30-day mortality, (4) and the feasibility of RV-FWLS in ICU by TTE was 74%. 

### 4.1. RV-FWLS and RV Dysfunction

RVsD is a common complication of COVID-19 (prevalence range 2% to 51%), increasing mortality, notably [2] especially when RVsD is associated with RV dilatation and acute cor pulmonale. Diagnosis of RVsD depends on the RV systolic parameters used. [2]. The predominance of longitudinal-oriented muscle fibers has led to the widespread use of RV systolic parameters based on RV-free wall longitudinal motion (TAPSE, RV-S, or RV-FWLS). In our study, the prevalence of RVsD was 40%, according to the recommended RV-FWLS cutoff value. One of the characteristics of COVID-19 is combined lung injuries (alveolar damage, interstitial edema, cell proliferation, fibrosis) and immunothrombosis (microvascular, deep venous thrombosis, arterial and pulmonary embolism) [19], which increase RV afterload, promoting RV dilatation and thus RV dysfunction. 

### 4.2. Performance of RV Parameters for Predicting 30-Day Mortality 

RV-FLWS is a promising and robust predictor of major cardiovascular events in several myocardial diseases [20]. At the beginning of the COVID-19 pandemic, the RV-FWLS seemed to be an attractive clinical tool for identifying patients with poor clinical outcomes, contrary to conventional parameters. In the Li study, conventional RV (TAPSE and RV-FAC) and RV-FWLS impairment, associated with gender and ARDS, were powerful predictors of mortality. The optimal cut-off for RV-FWL was −23%, 43.5% for RV-FAC, and 23 mm for TAPSE [8]. In our study, RV-FWLS and conventional parameters (TAPSE, RV-S’, and RV-FAC) failed to identify patients with an increased 30-day mortality. In a multicentric study of hospitalized COVID-19 patients, RV dysfunction defined by TAPSE < 17 mm, or an RV-S’ < 9.5 cm^−1^ or an RV-FAC < 35%, was not associated with hospital mortality [21]. At the bedside, the clinical interpretation of the cutoff value of RV-FWLS proposed by many authors is challenging because it is within the normal range in healthy subjects [8] or far away from the recommended threshold in cardiovascular disease [10]. 

### 4.3. RV-FWLS and 30-Day Mortality

We defined the RVsD from the recommended RV-FWLS value. However, we did not find any association between RV-FWLS and 30-day mortality, while others have previously reported such an association in COVID-19 patients [22]. However, our results are close to those of similar studies. In the Kim study, the median RV-FWLS was −22 [−27.2, −18.6]% and was not associated with COVID-19 disease severity in a logistic regression analysis [23]. In the Park study (n = 153 patients), RV-FWLS values were not different between survivors and non-survivors (−14.48 ± 5.63 vs. −14.77 ± 5.88, *p* = 0.88). In a multivariable COX analysis, RV-FWLS was not associated with mortality [10], contrary to age > 65 years old and the presence of an acute cor pulmonale.

We believe that our results can be explained by several pathophysiological factors encountered in COVID-19 pneumonia. First, COVID-19 pneumonia is frequently associated with RV dilatation or acute cor pulmonale (38% of patients in the RVsD group). Altered ventricular geometry and heterogeneity of myocardial contraction are well-known factors that can compromise the measured strain value [24,25]. Secondly, TTE was performed at an early stage of COVID-19, so patients with RVsD probably benefited from early medical optimization to avoid hemodynamic failure. As with other studies in this field, the variability of the timing of the echocardiography between the admission and examination for RV systolic function assessment was important [8,26]. It is unlikely that one single measurement at the initial stage of COVID-19 disease could accurately predict mortality in the complex intensive care setting. 

### 4.4. Feasibility of RV-FWLS in Critically Ill Patients

The main limitation of using RV-FLWS as a parameter in the clinical bedside routine to stratify a patient’s mortality risk is its feasibility. Although the TTE protocol was performed by level III operators with an automatic RV myocardial border delimitation and dedicated RV mode, 40 patients (26%) were excluded from our study. The Bleakly study assessed RV systolic function in critically ill COVID-19 patients, and the feasibility was 57% (49/90) [12]. This poor feasibility rate can be explained by the following factors: firstly, in the ICU, the adequate acoustic window is often impaired by mechanical ventilation, COVID-19 lung injuries (B lines, subpleural consolidation [27]), and suboptimal patient positioning. Secondly, RV-FWLS measurement requires an RV-focused four-chamber apical view, obtained with a more lateral transducer position than that used for a conventional apical-four chamber view [6], which is challenging in sedated and curarized patients. 

In conclusion, this lack of feasibility and the problematic interpretation of RV-FWLS cutoff values are probably why experts of the European Society of Intensive Care Medicine reported that RV strain does not add significant information value to the interpretation of RV function [3]. Even if RV-FWLS seems promising, its use in intensive care to stratify the risk of patient mortality does not seem relevant. 

### 4.5. Limitations

We acknowledge that the main limitations are the small sample size and the monocentric character of this study. However, our patient population is more significant than other studies on RV strain in COVID-19 patients [22]. Our study excluded patients under ECMO. Thus, it does not represent patients with the most severe ARDS. ECMO delivers a non-pulsatile blood flow and improves RV function by decreasing the RV afterload through a correction of the hypercapnia and the implementation of ultra-protective ventilation. These elements interact with RV diastolic and systolic function, making the interpretation of echocardiographic parameters challenging. We only analyzed RV-FWLS. For RV systolic assessment, RV longitudinal strain should be reported as the RV free wall deformation because the septal strain is considered as a part of left ventricular systolic function assessment. A catheterization study showed that RV-FWLS might better reflect RV systolic function than RV global longitudinal strain [28]. In addition, most studies assessing the cutoff and the prognostic values of longitudinal strain measured RV-FWLS [6]. The value of RV-FWLS depends on how it is measured. The software package, dedicated RV mode, determining the ROI (semi-automatically or automatically), and myocardial approach can make it challenging to compare RV strain results across studies [6]. As recommended, our study used an automated, endocardial approach on an RV-focused apical four-chamber view [6]. The RV-FWLS inter and intra-observer reproducibility assessed by ICC was >0.8, indicating good reliability [29]. 

## 5. Conclusions

In a cohort of COVID-19 pneumonia hospitalized in the ICU, RV-FWLS and RV conventional (TAPSE, RV-S’, and RV-FAC) parameters failed to predict 30-day mortality. The feasibility of RF-FWLS was 76% and RVsD defined by the RV-FWLS < 21% was not associated with 30-day mortality. The use of RV-FWLS in the ICU to stratify the mortality risk of COVID-19 pneumonia did not seem relevant. Further studies with a larger sample size are mandatory to confirm our findings. 

## Figures and Tables

**Figure 1 jcm-11-03629-f001:**
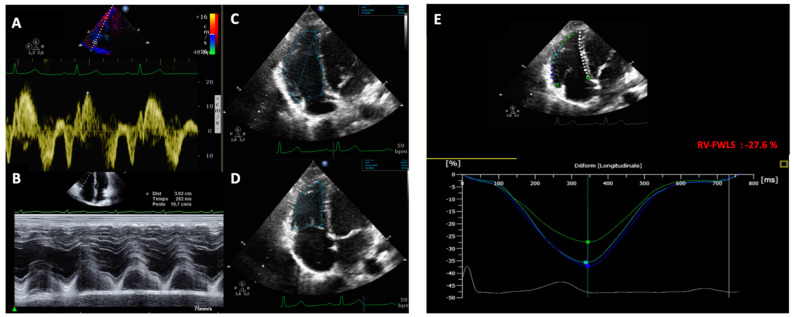
Measurement of the RV conventional parameters and the RV-FWLS in a RV-focused apical four-chamber view. (**A**) RV-S’ wave measured in tissular Doppler imaging. (**B**) TAPSE measure in M-mode. (**C**,**D**) End-diastolic RV area and end-systolic RV area measured for RV-FAC. (**E**) Automatic RV-FWLS measured by the QLAB AutoStrain Software.

**Figure 2 jcm-11-03629-f002:**
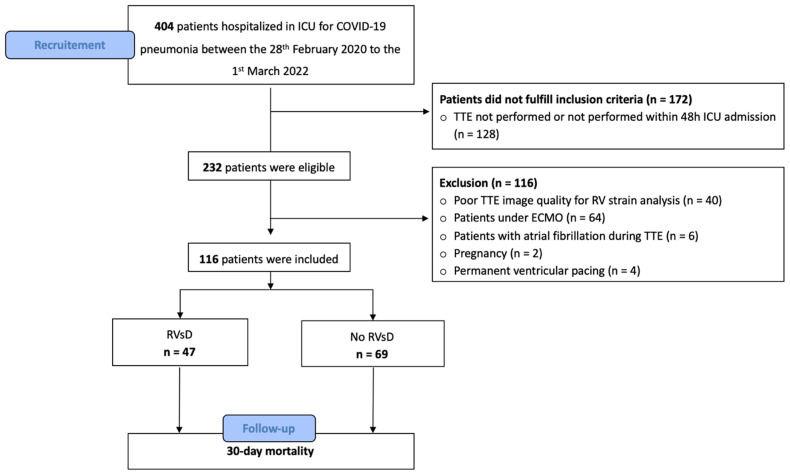
Flow chart of the study.

**Figure 3 jcm-11-03629-f003:**
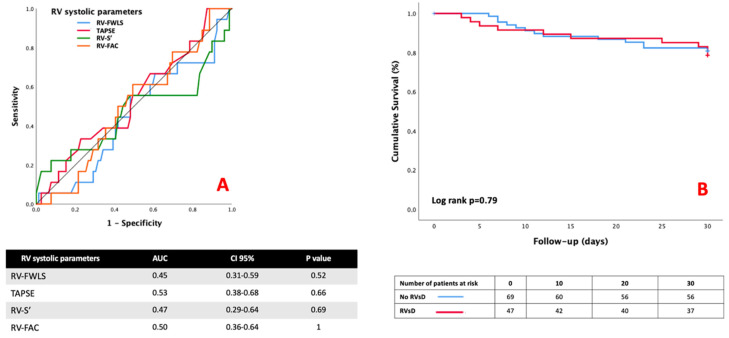
(**A**) ROC curve analysis of RV systolic parameters for predicting 30-day mortality. (**B**) Kaplan–Meier curve analysis between the RVsD group and no-RVsD group.

**Table 1 jcm-11-03629-t001:** Demographics, biological and computed tomography data before TTE.

Variables	No RVsD (n = 69)	RVsD (n = 47)	*p*
Age (years)	60 (59–68)	62 (59–73)	0.39
BMI (kg m^−2^)	30.4 (25.7–34.1)	30.2 (26.6–34.9)	0.67
Male gender (n; %)	45 (65)	33 (70)	0.68
SAPS II score	32 (20–56)	39 (24–61)	0.29
Medical history			
No history	10 (14)	7 (15)	0.99
Hypertension	34 (49)	24 (51)	0.99
Diabetes	16 (23)	14 (30)	0.51
Dyslipidemia	18 (26)	18 (38)	0.22
Smoking (former or active)	12 (17)	10 (21)	0.63
Chronic renal disease	5 (7)	5 (10	0.52
COPD/asthma	16 (23)	4 (9)	0.08
Coronary or peripheral artery disease	4 (6)	7 (15)	0.17
Valvular heart disease	2 (3)	5 (10)	0.12
CT scan (n = 107/116)	62 (89)	45 (95)	
Frosted glass	59 (95)	44 (97)	0.63
Consilodation	37 (62)	27 (64)	0.99
Crazy Paving	19 (32)	9 (22)	0.36
Lung involvement > 50%	30 (43)	22 (46)	0.99
Pulmonary embolism	1 (2)	4 (9)	0.15
Biological data before TTE			
Lactate (mmol^−1^)	1.7 (1.3–2.1)	1.9 (1.3–2.3)	0.16
Serum creatinine (µmol L^−1^)	66 (54–88)	70 (54–88)	0.61
BNP (pg mL^−1^)	59 (31–101)	79 (34–243)	0.12
Troponin Tc HS (ng mL^−1^)	16 (6–36)	27 (8–95)	0.06
Procalcitonin (µg L^−1^)	0.40 (0.18–0.75)	0.57 (0.13–1.60)	0.47
C reactive protein (mg L^−1^)	150 (96–215)	160 (84–268)	0.62
Time to first symptom to ICU admission (days)	8 (5–10)	8 (6–12)	0.44
Period of hospitalization in ICU *			
- Period A	16 (23)	9 (19)	0.65
- Period B	13 (19)	6 (13)	0.45
- Period C	40 (58)	32 (68)	0.33

Data are presented as median (interquartile range)) and number (percentage). BMI: body mass index. BNP: brain natriuretic peptide; CT: computerized tomography. COPD: chronic obstructive pulmonary disease; SAPS: simplified acute physiology score. TTE: transthoracic echocardiography. * period A (between 28 February 2020 and 1 June 2020), period B (between 1 September 2020 and 15 April 2020) and period C (between 12 December 2020 and 15 March 2022).

**Table 2 jcm-11-03629-t002:** Hemodynamics, TEE parameters and outcomes.

	No RVsD (n = 69)	RVsD (n = 47)	*p*
Hemodynamic parameters during TTE			
Heart rate (bpm)	82 (74–92)	87 (74–102)	0.14
Systolic arterial pressure (mmHg)	130 (114–144)	127 (113–146)	0.71
Mean arterial pressure (mmHg)	86 (73–97)	86 (80–95)	0.53
Diastolic arterial pressure (mmHg)	70 (58–80)	69 (61–79)	0.44
Mechanical ventilation, n (%)	37 (54)	32 (70)	0.12
PaO_2_ (mmHg)	76 (65–92)	80 (71–107)	0.12
Vasopressor use, n (%)	19 (27)	13 (28)	0.99
TTE parameters			
RV basal dimension (mm)	45 (40–51)	44 (38–50)	0.21
RV mid-cavity dimension (mm)	38 (29–43)	37 (32–41)	0.54
RV longitudinal dimension (mm)	77 (69–85)	76 (70–84)	0.83
RV EDA (cm^2^)	21 (16–26)	21 (17–24)	0.94
RV ESA (cm^2^)	11 (7–14)	11 (9.5–15)	0.23
RV EDA/LV EDA	0.68 (0.55–0.92)	0.67 (0.57–0.87)	0.91
Acute cor pulmonale, n (%)	18 (26)	18 (38)	0.22
RA volume (mL)	33 (27–35)	32 (27–40)	0.63
RA area (cm^2^)	14.6 (9.6–19.2)	14.6 (7.1–20.5)	0.61
Left ventricular ejection fraction (%)	64 (53–70)	63 (53–73)	0.57
RV Systolic Function Parameters			
TAPSE (mm)	24 (21–27)	21 (19–25)	0.03
RV-S’ (cm s^−1^)	16 (13–19)	16 (13–19)	0.91
RV-FAC (%)	47 (41–54)	41 (34–47)	0.006
RV-FWLS (%)	26.7 (24.1–30.1)	17.5 (15.3–19.0)	0.0001
Outcomes (n,%)			
Ventilator acquired pneumonia	34 (49)	28 (61)	0.56
Renal replacement therapy	10 (14)	7 (15)	0.99
Pulmonary embolism	4 (6)	5 (11)	0.48
Cardiogenic shock	3 (4)	6 (13)	0.15
Veno-venous ECMO	8 (12)	4 (9)	0.76
Tracheotomy	10 (15)	3 (7)	0.23
Time under mechanical ventilation (days)	18 (11–30)	18 (10–27)	0.47
30-day mortality	13 (19)	10 (21)	0.82
ICU mortality	17 (25)	10 (21)	0.81
ICU stay (days)	11 (6–29)	17 (7–31)	0.27

Data are presented as median (interquartile range) and number (percentage). EDA: end-diastolic area; ECMO: extracorporeal membrane oxygenation; ESA: end-systolic area; ICU: intensive care unit. RV: right ventricle; RA: right atrial; RV-FAC: right ventricle fractional area change; RV-FWLS: right ventricle free wall longitudinal strain; RV-S’: right ventricle S’ wave; TAPSE: tricuspid annular plane systolic excursion; TTE: transthoracic echocardiography.

**Table 3 jcm-11-03629-t003:** Univariate and multivariate Cox analysis of variables associated with 30-day mortality.

Variables	30 Days Mortality			
	Univariate Analysis		Multivariate Analysis	
	HR (95%CI)	*p*	HR (95%CI)	*p*
SAPS II (for each point)	1.03 (1.01–1.05)	0.02	1.04 (1.01–1.06)	0.005
Age > 65 years old	5.01 (2.10–12.37)	0.001	7.51 (2.63–21.44)	0.0001
Mechanical ventilation *	1.41 (0.57–3.46)	0.45	-	-
RV-FWLS < 21%	1.12 (0.49–2.55)	0.78	-	-
Acute cor pulmonale *	3.48 (1.48–8.14)	0.004	7.53 (2.58–21.9)	0.0001
Pulmonary embolism before TTE	0.91 (0.12–6.77)	0.92	-	-
SOFA cv *	1.47 (0.61–3.51)	0.38	-	-
Period of inclusion				
- Period A	0.54 (0.24–2.11)	0.72	-	-
- Period B	0.72 (0.21–2.41)	0.59	-	-
- Period C	1.15 (0.62–3.68)	0.36	-	-

* Factors recorded during TTE exam. CI: confidence interval; CV: cardiovascular; HR: hazard ratio; SAPS: simplified acute physiology score; SOFA: sepsis organ failure assessment; RV: right ventricle; RV-FWLS: right ventricle free wall longitudinal strain; TTE: transthoracic echocardiography.

## Data Availability

The datasets used and/or analyzed during the current study are available from the corresponding author on reasonable request.

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
