# Peer review of "Feasibility, Prediction and Association of Right Ventricular Free Wall Longitudinal Strain with 30-Day Mortality in Severe COVID-19 Pneumonia: A Prospective Study"

_jcm, 2022, doi:10.3390/jcm11133629_

Round 1

Reviewer 1 Report

This article by Beyls et al presents the authors experience with RV dysfunction in COVID-19 patients hospitalized in an ICU setting. Echocardiographic studies were performed to address the possible association between several markers of RV systolic dysfunction and 30-day mortality, notably including right ventricular free wall longitudinal strain (RV-FWLS). Although 40% of patents fulfilled criteria for RV dysfunction, none of the measured parameters, and most especially RV-FWLS, showed any significant association with mortality.

This is a well-conducted and interesting study, which adds information to the field of RV dysfunction and its echocardiographic assessment in severe COVID-19 patients. I have only some minor comments regarding mostly spelling and grammatical issues, that I propose the authors to amend in a revised version of the manuscript.  

Minor comments

1) Page 3 : Please change « Figure X » to Figure 1.

2) In the legend of Figure 1, measurment should be written measurement.

3) In figure 2: A) “Recrutement” should be written recruitment. B) “Patients did not fulfilled” should be written patients did not fulfill. C) “Women pregnancy” should be simply written pregnancy.

4) Table 1. A) “Creatinine serique” should be written: serum creatinine. (B) “Troponine” should be written troponin; C) “Hospitalizarion” should be written hospitalization.

5) Table 2. “Hear Rate” should be written Heart Rate.

6) Page 6. In the sentence: “RV-FWLS and conventional RV parameters were interred into a ROC analysis”, I propose to use another term than “interred” which is not appropriate here.

7) Page 7. “lok rank test” should be written log rank test.

8) Page 8: “Feasibility of RV-FWLS in critical ill” should be written: Feasibility of RV-FWLS in critically ill patients

9) Page 9: “ultra-protect ventilation” should be written: ultra-protective ventilation

10) Page 9. Discussion: “ECMO allows the improvement of RV function by decreasing the RV afterload through a correction of the hypercapnia and the implementation of ultra-protect ventilation. Several mechanisms that interact with RV function make it difficult not only the assessment of RV function but also the comparison it with patients without ECMO. ECMO allows RV unloading by providing ultra-protective ventilation and delivering additional non-pulsatile blood flow”. These sentences are a bit redundant, and also include some grammatical mistakes. Please amend this part of the discussion.

11) Please cite the following reference, which is a recent review on the topics of RV dysfunction in COVID-19: Bonnemain J, Ltaief Z, Liaudet L. The Right Ventricle in COVID-19. J Clin Med. 2021 Jun 8;10(12):2535. doi: 10.3390/jcm10122535. PMID: 34200990; PMCID: PMC8230058.

Author Response

Reviewer 1

This article by Beyls et al presents the authors experience with RV dysfunction in COVID-19 patients hospitalized in an ICU setting. Echocardiographic studies were performed to address the possible association between several markers of RV systolic dysfunction and 30-day mortality, notably including right ventricular free wall longitudinal strain (RV-FWLS). Although 40% of patents fulfilled criteria for RV dysfunction, none of the measured parameters, and most especially RV-FWLS, showed any significant association with mortality.

This is a well-conducted and interesting study, which adds information to the field of RV dysfunction and its echocardiographic assessment in severe COVID-19 patients. I have only some minor comments regarding mostly spelling and grammatical issues, that I propose the authors to amend in a revised version of the manuscript.  

Minor comments

1) Page 3 : Please change « Figure X » to Figure 1.

2) In the legend of Figure 1, measurement should be written measurement.

3) In figure 2: A) “Recrutement” should be written recruitment. B) “Patients did not fulfilled” should be written patients did not fulfill. C) “Women pregnancy” should be simply written pregnancy.

4) Table 1. A) “Creatinine serique” should be written: serum creatinine. (B) “Troponine” should be written troponin; C) “Hospitalizarion” should be written hospitalization.

5) Table 2. “Hear Rate” should be written Heart Rate.

6) Page 6. In the sentence: “RV-FWLS and conventional RV parameters were interred into a ROC analysis”, I propose to use another term than “interred” which is not appropriate here.

7) Page 7. “lok rank test” should be written log rank test.

8) Page 8: “Feasibility of RV-FWLS in critical ill” should be written: Feasibility of RV-FWLS in critically ill patients

9) Page 9: “ultra-protect ventilation” should be written: ultra-protective ventilation.

Response: We thank the reviewer for his remarks. We corrected them and we apologize for these mistakes.

10) Page 9. Discussion: “ECMO allows the improvement of RV function by decreasing the RV afterload through a correction of the hypercapnia and the implementation of ultra-protect ventilation. Several mechanisms that interact with RV function make it difficult not only the assessment of RV function but also the comparison it with patients without ECMO. ECMO allows RV unloading by providing ultra-protective ventilation and delivering additional non-pulsatile blood flow”. These sentences are a bit redundant, and also include some grammatical mistakes. Please amend this part of the discussion.

Response: We thank the reviewer for this remark. We change the sentence for more clarity, page 10, line 359, as follows: ECMO delivers a non-pulsatile blood flow and improves RV function by decreasing the RV afterload through a correction of the hypercapnia and the implementation of ultra-protective ventilation. These elements interact with RV diastolic and systolic function, making the interpretation of echocardiographic parameters challenging.

11) Please cite the following reference, which is a recent review on the topics of RV dysfunction in COVID-19: Bonnemain J, Ltaief Z, Liaudet L. The Right Ventricle in COVID-19. J Clin Med. 2021 Jun 8;10(12):2535. doi: 10.3390/jcm10122535. PMID: 34200990; PMCID: PMC8230058.

Response: As recommended, we add the reference on page 1, line 37

Reviewer 2 Report

I read with high interest the manuscript of "Feasibility, Prediction and Association of Right Ventricular Free Wall Longitudinal Strain with 30-Mortality in Severe COVID-19 Pneumonia: A Prospective Study" by Christophe Beyls et al.

With their research, the authors try to elucidate whether parameters of right ventricular (RV) function, especially RV free wall longitudinal strain (RV-FWLS) can serve as predictors of mortality in a cohort of severly ill patients suffering from COVID-19 pneumonia. They come to the conclusion, that this is not the case. Although other research had already suggested potential of these parameters to predict mortality, Beyls et al. argue that their analysis might be more robust, since it focuses on a collective that can be better assessed since it has less confounders, such as ECMO.

The authors should be commended on their work, since it addresses a very important area of critical care medicine, with COVID-19 still plaguing the world. 

The paper is well written, and there are only minor issues that I think should be addressed:

Spelling:
- "ICU" had already be defined before, hence it is redundant in the sentence "Adult patients (>18 years of age) admitted to our intensive care unit (ICU) for docu- mented severe COVID-19 pneumonia"

- "women’s pregnancy" seems redundant

- There is a ")" missing in "he predominance of longitudinal-oriented muscle fibers has led to the widespread use of RV systolic parameters based on RV-free wall longitudinal motion (TAPSE, RV-S, or RV-FWLS."

Content

- RA volume was assessed using the Simpson's method for volume calculation. Strictly speaking, this makes no sense, since this would need at least a second plane. I know that this is done regularly to compare to the LA - here comes the second problem: the method was invented only for LV assessment as it has the "bullet shape" that can be measured using the Simpson's measurement. At least, also add the values for the RA area that are readily provided within the same measurements. 

- It says "Figure X" when it probably should say "figure 1" under 2.M&M, TTE. I further suggest to move the reference to the figure and form it as a standalone sentence. At the moment, it suggests the reader would find more info on strain alone, when the figure shows all RV assessments. Maybe something simple like "Figure 1 offers information on RV function assessment" after the sentence "All operators had a level III compe- tence of general adult TTE [15]." could introduce to the details following. Just a thought. Also, in the figure, the panels are indicated in red color. I suggest to use white, because  - at least not in my copy - "A" and "B" become invisible when printed. Finally, in the description, please be specific on C and D, so that all reads know which is which.

- in the tables, there are numerous p-values showing 1. This is impossible, but easily explained as the rounding error of SPSS and also easily fixed by typing 0.99.

- I am unsure about the sentence "Secondly, RV-FWLS measurement requires a focus RV four-chamber apical view, obtained with a more lateral transducer position than the one used for conventional apical-four chamber view [5] which is challenging in sedated and curarized patients.": while the general statement is true, that the focused RV view is harder to reach in this patient population, I am quite certain that you have to go more medial in comparison to the A4CV to focus on that view (the RV is also more medial than the LV).

- Please remove "6. patents"

Author Response

Reviewer 2

I read with high interest the manuscript of "Feasibility, Prediction and Association of Right Ventricular Free Wall Longitudinal Strain with 30-Mortality in Severe COVID-19 Pneumonia: A Prospective Study" by Christophe Beyls et al.

With their research, the authors try to elucidate whether parameters of right ventricular (RV) function, especially RV free wall longitudinal strain (RV-FWLS) can serve as predictors of mortality in a cohort of severly ill patients suffering from COVID-19 pneumonia. They come to the conclusion, that this is not the case. Although other research had already suggested potential of these parameters to predict mortality, Beyls et al. argue that their analysis might be more robust, since it focuses on a collective that can be better assessed since it has less confounders, such as ECMO.

The authors should be commended on their work, since it addresses a very important area of critical care medicine, with COVID-19 still plaguing the world. 

The paper is well written, and there are only minor issues that I think should be addressed:

Spelling:
- "ICU" had already be defined before, hence it is redundant in the sentence "Adult patients (>18 years of age) admitted to our intensive care unit (ICU) for docu- mented severe COVID-19 pneumonia"

Response: We thank the reviewer for this remark. We corrected the sentence page 2, line 67, as follows: Adult patients admitted to our ICU for documented severe COVID-19 pneumonia

- "women’s pregnancy" seems redundant.

- There is a ")" missing in "he predominance of longitudinal-oriented muscle fibers has led to the widespread use of RV systolic parameters based on RV-free wall longitudinal motion (TAPSE, RV-S, or RV-FWLS."

Response: We thank the reviewer for this remark and we corrected them.

Content

- RA volume was assessed using the Simpson's method for volume calculation. Strictly speaking, this makes no sense, since this would need at least a second plane. I know that this is done regularly to compare to the LA - here comes the second problem: the method was invented only for LV assessment as it has the "bullet shape" that can be measured using the Simpson's measurement. At least, also add the values for the RA area that are readily provided within the same measurements. 

Response: We thank the reviewer for this remark. The RA volume was assessed by the disks summation technique, as recommended in ASE guidelines. We agree that this technique that it is not the most reliable technique and that a 3D acquisition would be more appreciable. As recommended we ad in the table 2 the RA area.

- It says "Figure X" when it probably should say "figure 1" under 2.M&M, TTE. I further suggest to move the reference to the figure and form it as a standalone sentence. At the moment, it suggests the reader would find more info on strain alone, when the figure shows all RV assessments. Maybe something simple like "Figure 1 offers information on RV function assessment" after the sentence "All operators had a level III compe- tence of general adult TTE [15]." could introduce to the details following. Just a thought. Also, in the figure, the panels are indicated in red color. I suggest to use white, because  - at least not in my copy - "A" and "B" become invisible when printed. Finally, in the description, please be specific on C and D, so that all reads know which is which.

Response: We thank the reviewer for this remark. We complete the method section for introduce the Figure 1, page 3, line 104, as follows: RV conventionnal and 2D-strain parameters are summarized in Figure 1 and we change the color of the Figure 1 letter as suggested.

- in the tables, there are numerous p-values showing 1. This is impossible, but easily explained as the rounding error of SPSS and also easily fixed by typing 0.99.

Response: We thank the reviewer for this remark and we corrected the p-values in the table.

- I am unsure about the sentence "Secondly, RV-FWLS measurement requires a focus RV four-chamber apical view, obtained with a more lateral transducer position than the one used for conventional apical-four chamber view [5] which is challenging in sedated and curarized patients.": while the general statement is true, that the focused RV view is harder to reach in this patient population, I am quite certain that you have to go more medial in comparison to the A4CV to focus on that view (the RV is also more medial than the LV).

Response: We thank the reviewer for this remark. Indeed, the anatomical position of the RV would like us to approach the midline to improve its exploration in echocardiography, however, a lateral approach allows us to clear all the free wall of the right ventricle and thus avoid a foreshortening A4CV. (https://doi.org/10.1093/ehjci/jeac022).

- Please remove "6. patents"

Response: As recommended, we remove this part.  

Reviewer 3 Report

Comments

Dr.Beyls and colleagues submitted the manuscript titled " Feasibility, Prediction and Association of Right Ventricular Free Wall Longitudinal Strain with 30-Mortality in Severe COVID-19 Pneumonia: A Prospective Study”. This study explored the feasibility of RV-FWLS to predict 30-day mortality and the association between RV-FWLS and 30- day mortality. The main conclusion was that RV-FWLS failed to predict 30-day mortality, there was no association between RV-FWLS and 30-day mortality.

1.      Right ventricular dysfunction often leads to systemic congestion, such as liver congestion, gastrointestinal congestion, lower limb edema, ascites, pleural effusion, etc. But it will not aggravate the common causes of death in patients with covid-19, including embolism, cytokine storm and so on. Therefore, it is difficult to establish a relationship between right ventricular dysfunction and mortality in patients with covid-19.

2.      Sample size is small.

3.      There were no positive results. And the study did not solve specific problems or provide reference value for solving specific problems.

4.      The discussion in this paper is inconsistent with the results, “feasibility of RF-FWLS” should be 74%, but it is wrongly written in the discussion as 76%

Author Response

Reviewer 3

Dr.Beyls and colleagues submitted the manuscript titled " Feasibility, Prediction and Association of Right Ventricular Free Wall Longitudinal Strain with 30-Mortality in Severe COVID-19 Pneumonia: A Prospective Study”. This study explored the feasibility of RV-FWLS to predict 30-day mortality and the association between RV-FWLS and 30- day mortality. The main conclusion was that RV-FWLS failed to predict 30-day mortality, there was no association between RV-FWLS and 30-day mortality.

  1. Right ventricular dysfunction often leads to systemic congestion, such as liver congestion, gastrointestinal congestion, lower limb edema, ascites, pleural effusion, etc. But it will not aggravate the common causes of death in patients with covid-19, including embolism, cytokine storm and so on. Therefore, it is difficult to establish a relationship between right ventricular dysfunction and mortality in patients with covid-19.

Response: We thank the reviewer for this remark. Indeed, RV dysfunction can lead to organ dysfunction by congestion and hemodynamic instability. Several studies and meta-analyses have shown that RV dysfunction was associated with increased mortality risk for COVID-19 patients. However, as we try to offer, it depends on the technique (echography, MRI, clinical score) and the parameters used to define the RV dysfunction. We also agree that the diagnosis of RV dysfunction alone can predict a patient's mortality. We discuss this point on page 10, line 331: It is unlikely that one single measurement at the initial stage of COVID-19 disease could accurately predict mortality in the complex intensive care setting.

  1. Sample size is small.

Response: We thank the reviewer for his remark. We agree with the reviewer that the sample size of our study is limited and requires larger multicentric studies to confirm our findings. However, our patient population is more significant than other studies evaluating RV strain in COVID-19 disease. We discuss this point in the limits section, page 11, line 356: We acknowledge that the main limitations are the small sample size and the monocentric character of this study. However, our patient population is more significant than other studies on RV strain in COVID-19 patients [22].

  1. There were no positive results. And the study did not solve specific problems or provide a reference value for solving specific problems.

Response: We thank the reviewer for this remark. Indeed, our study does not find an association between RV dysfunction defined by the RV-FWLS and 30-days mortality. This finding is part of a current debate about the appropriateness of RV strain in critically ill patients. Given this result, we think that stratifying the risk of patient mortality based on RV-strain, as some authors suggest, seems not relevant.

  1. The discussion in this paper is inconsistent with the results, “feasibility of RF-FWLS” should be 74%, but it is wrongly written in the discussion as 76%

Response: We thank the reviewer for this remark. We correct the sentence on page 9, line 280, and we apologize for this error.
